# An andesitic source for Jack Hills zircon supports onset of plate tectonics in the Hadean

Simon Turner [1✉], Simon Wilde [2], Gerhard Wörner [3], Bruce Schaefer[1] & Yi-Jen Lai[1]

The composition and origin of Earth's early crust remains hotly debated. Here we use partition coefficients to invert the trace element composition of 4.3–3.3 Gyr Jack Hills zircons to calculate the composition of the melts from which they crystallised. Using this approach, the average $SiO_2$ content of these melts was 59 ± 6 wt. % with Th/Nb, Dy/Yb and Sr/Y ratios of 2.7 ± 1.9, 0.9 ± 0.2 and 1.6 ± 0.7, respectively. Such features strongly indicate that the protolith for the Jack Hills zircons was not an intra-plate mafic rock, nor a TTG (tondjhemite-tonalite-granodiorite) or a Sudbury-like impact melt. Instead, the inferred equilibrium melts are much more similar to andesites formed in modern subduction settings. We find no evidence for any secular variation between 4.3 and 3.3 Gyr implying little change in the composition or tectonic affinity of the Earth's early crust from the Hadean to Mesoarchaean.

[1] Department of Earth and Planetary Sciences, Macquarie University, Sydney, NSW 2109, Australia. [2] Department of Applied Geology, Curtin University, PO Box U1987, Perth, WA 6845, Australia. [3] Abteilung Geochemie, Geowissenschaftliches Zentrum Göttingen (GZG), 37077 Göttingen, Germany. ✉email: simon.turner@mq.edu.au

The composition and tectonic affiliation of the Earths earliest crust are two of the most fundamental questions in Earth science[1–4] but challenging to constrain due to the lack of suitably aged rocks. Many studies have concentrated on 4.4–3.3 Gyr old aged zircons extracted from metasediments in the Jack Hills in Western Australia[1,5–9] but consensus on the nature of their protolith(s) remains elusive. Some inferences about the bulk composition of their protoliths have come from the Hf isotope composition of the zircons[5,8,9]. One thing that is not in dispute is that these zircons derive from igneous source rocks[7,10,11]. Most crystallised at temperatures around 700 ± 50 °C[12,13] but the parental magmas would likely have been hotter. It is generally accepted that elevated U/Th ratios (>0.1) indicate that the zircons derive from igneous source rocks[7,10,11] but thereafter opinions differ. Some advocate low temperature, hydrous granites as the protoliths, implying conditions similar to those on Earth today[14,15]. Others infer more mafic, possibly intra-plate basaltic source rocks[9,16] or even that the zircons crystallised from impact melts[17–19]. The latter interpretations do not require plate tectonics. Other studies have suggested that plate tectonics was in a stop—start mode on the early Earth[3,20] which would predict that key geochemical signatures (e.g., $SiO_2$ content, Th/Nb ratios) alternate between within-plate and subduction like over time during the Archaean. As many of these conclusions are based on the Hf isotope signature of the zircons[1,5,8,9] there is a need for a complementary approach. Whilst mostly concentrating on rare earth element (REE) abundances, zircon trace element data have been collected in several different studies[11,21] and fingerprinting of these also indicate the Jack Hills zircons most likely crystallised from continental igneous rocks rather than fractionates of mafic intra-plate melts[5,21]. However, additional information can be extracted by exploiting a wider range of trace element data from the Jack Hills zircons. These suggest that they crystallised from melts with trace element signatures akin to modern arc andesites making an early onset of modern style plate tectonics entirely permissible.

## Results

We have prepared a new aliquot of zircons from Jack Hills metasediment and obtained their U-Pb ages using standard SHRIMP methodology (see Methods section and Supplementary Data 1 and Supplementary Data 2). The ages range from 4.3 to 3.3 Ga consistent with earlier work[5–7]. We then used a standard laser-extraction technique (LA-ICP-MS) to obtain trace element data from the same location from which the ages were obtained (see Methods section and Supplementary Table S2). Next, we used zircon/melt partition coefficients to calculate the trace element composition of the melts from which the zircons crystallised (i.e., their protoliths). For this purpose the choice of partition coefficients is critical. The majority of studies do not report large ion lithophile and high field strength element data that are especially critical here. Accordingly, we have used LA-ICP-MS partition coefficients from the only experimental study that analysed both a wide range of trace elements and that also showed these to be consistent with the lattice strain model[22]. These experiments were conducted on an andesitic composition that, as we demonstrate below seem highly applicable to the Jack Hills zircons. Moreover, although a temperature dependence on absolute partition coefficient values has been well demonstrated, the ratios of partition coefficient (as we use throughout) are far less prone to this problem[22,23]. Moreover, zircons from plutonic rocks often contain micro-inclusions and their hosts are notoriously heterogeneous, rendering most empirical partition coefficients significantly inaccurate.

The new results are illustrated on a series of time-sequence diagrams in Fig. 1. Th/Y ratios were used to calculate the $SiO_2$ content of the equilibrium melts (see Methods section) and these range from 51 to 68 wt.% with an average of 59 wt.% (Fig. 1a). These overlap estimates for the present day continental crust that has a broadly andesitic composition[24]. Th/Nb ratios provide one measure of the presence or absence of a negative Nb anomaly that is absent in mid-ocean ridge and intra-plate rocks. The melt Th/Nb ratios we have estimated from the Jack Hills zircons are 2.7 ± 1.9, significantly higher than any modern oceanic rocks and extend from values akin to typical continental crust to even higher values (Fig. 1b). Finally, we also calculated the temperatures at which these zircons crystallised yielding 700 ± 50 °C, consistent with previous observations[12,13]. However, zircon is typically a late crystallised phase in magmatic systems and so these must be regarded as minimum magmatic temperatures if the melts from which the zircons formed evolved from more mafic parental melts. In respect to this it is interesting that the two oldest zircons appear to record higher crystallisation temperatures than the remaining zircons. A key observation is that

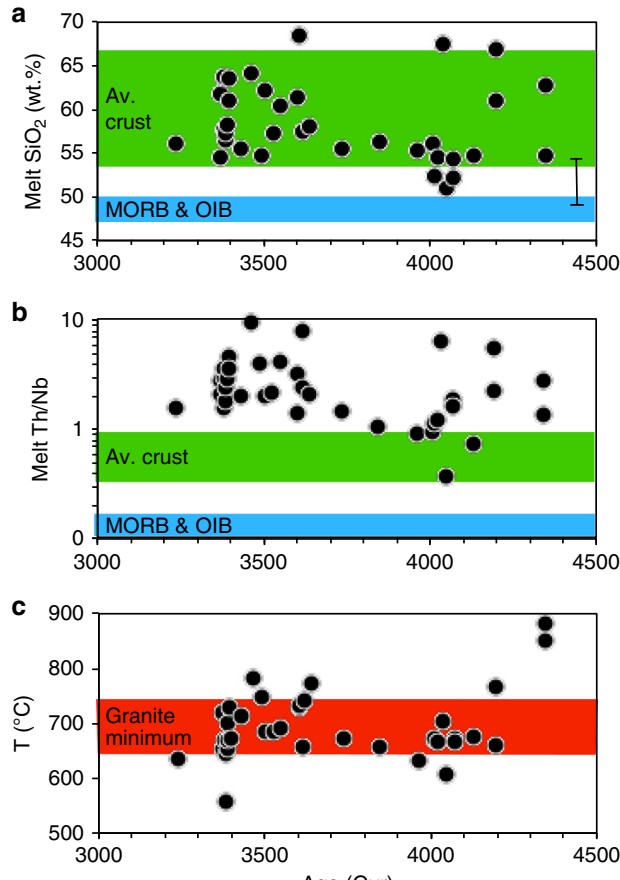

**Fig. 1 Compositional evolution of Jack Hills zircon protoliths through time. a** Melt $SiO_2$ calculated from inferred melt Th/Y is similar to average bulk crust. **b** Melt Th/Nb ratios are distinct from mid-ocean ridge and ocean island basalts but similar to or higher than average crust. **c** Zircon crystallisation temperatures are similar to the granite minimum though the parental magmas would have been hotter. Errors are smaller than the symbol size. There is no secular change in any of these parameters and all are features of typical subduction zone andesites. Average compositions of continental crustal[24] and MORB and OIB[43] are also shown. Error bar in **a** is 1σ (see Methods section).

neither the calculated melt $SiO_2$ contents and Th/Nb ratios or the zircon crystallisation temperatures show any statistically meaningful ($R^2 > 0.5$) secular change from the Hadean to Mesoarchaean (i.e., between 4.3 and 3.3 Ga).

## Discussion

The combination of elevated $SiO_2$ contents and elevated Th/Nb ratios are characteristics of andesitic melts formed in a modern subduction setting. Although some remain sceptical, plots like the Th/Yb versus Nb/Yb diagram shown in Fig. 2a have been repeatedly appraised to help distinguish the tectonic setting in which ancient rocks formed[25,26]. As can be seen the protoliths of the Jack Hills zircons clearly plot well above the fields for oceanic rocks and in or above the field for modern arc rocks[3]. Trondhjemite-tonalite-granodiorite suites (TTGs) characterise much of the felsic portion of the Archaean geological record and might be considered an obvious potential candidate source for the Jack Hills zircons. Many TTGs have elevated Th/Nb (and Sr/Y) ratios and that has sparked much debate as to whether these rock associations formed via subduction[26]. However, the Jack Hills protolihts do not overlap the field of TTGs on Fig. 2a. They are also distinct from the composition of rocks from Sudbury that are inferred to reflect crystallisation from an impact-derived melt (Fig. 2a).

A caveat to the preceding discussion is the choice of zircon/ melt partition coefficients as noted earlier. In order to estimate the effect of differing partition coefficients we also applied the partition coefficients from a recent empirical study of a high silica rhyolite glass[27]. These should avoid many of the problems with whole rock estimates and were also shown to be consistent with the lattice strain model. Unlike Th and the rare earth elements (REE), Nb is especially vulnerable to the choice of zircon/melt partition coefficient because it is quite incompatible in zircon and there can be analytical difficulties in measuring Nb[22]. For example, Nb is significantly more incompatible in the empirical partitioning data[27]. We note that the latter were obtained from a high silica rhyolite whereas the broadly andesitic $SiO_2$ content inferred for the Jack Hills zircon protoliths is more consistent with the andesitic starting composition used in the experimental study[22]. Nevertheless, if we use the empirical values[27] the calculated melts in equilibrium with the Jack Hills zircons shift to higher Nb/Yb such that they can overlap the TTG field on Fig. 2a. Irrespective, they remain displaced well above the fields for oceanic rocks (i.e., at elevated Th/Nb).

Another well-established feature of the TTGs is that they have steep heavy REE patterns and elevated Sr/Y ratios (>10) indicative of an important role for garnet either during partial melting or crystal fractionation[27]. REE partitioning into garnet is well constrained and ratios of the partition coefficients for the heavy REE (which are highly compatible in zircon) appear particularly robust[22,28]. Thus, a second striking result of our investigation is that the inferred Jack Hills protoliths had flat REE patterns and so their Dy/Yb ratios do not overlap the TTG field on Fig. 2b and they have an average Sr/Y ratio of only 1.6 ± 0.7 (not shown). Their Dy/Yb ratios are also distinct from the composition of rocks from the Sudbury intrusion (Fig. 2b).

What then, can be said about the nature of the Jack Hills zircon protoliths? The most robust conclusions are that they had intermediate $SiO_2$ contents and elevated Th/Nb ratios coupled with low Dy/Yb ratios. These results do not change even if a very different set of partition coefficients are used (see Fig. 2). Since impact melts have the composition of their target rock it is unsurprising that the Sudbury rocks lie close to average continental crust (Fig. 2b). As shown on Fig. 2b, the melts in equilibrium with the Jack Hills zircons are most similar to modern

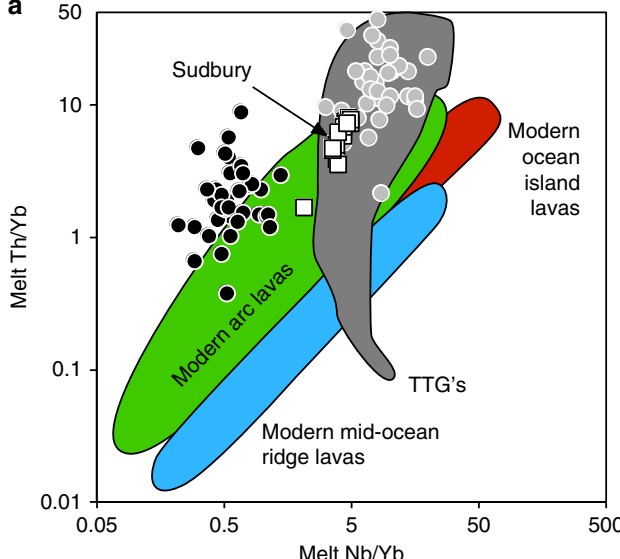

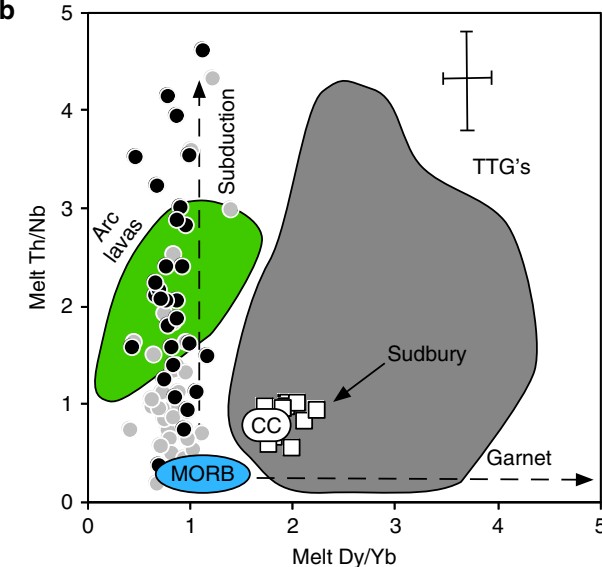

**Fig. 2 Tectonic affinity of the Jack Hills zircon protoliths. a** Trace element discrimination diagram[25] showing comparison of melts in equilibrium with Jack Hills zircons with the fields for modern arc rocks, mid-ocean ridge and ocean island basalts. **b** Plot of Th/Nb versus Dy/Yb for melts in equilibrium with the Jack Hills zircons along with vectors indicating the contrasting effects of subduction and melting (or crystallisation) in the presence of garnet. Black circles assume experimental zircon/melt partition coefficients[22]. For comparison, the grey circles assume empirical zircon/ melt partition coefficients[28]. Also shown are the composition of Sudbury rocks (white squares) inferred to have formed from impact-induced melting[44] and the TTG field for trondhjemite-tonalite-granodiorite suites (data compilation from GEOROC). Average crustal composition[24] (CC) and MORB[43] are also shown. Error bars in **b** are 1σ (see Methods section).

day arc lavas. Interestingly the average composition of the inferred Jack Hills protolith is very similar to the average low-Ti enriched basaltic andesites from the Nuvvuagittuq Greenstone Belt ($SiO_2 \sim 55$ wt.%; Th/Nb ~ 0.8) that may be 3.8 or 4.3 Gyr in age[29] and have been argued to derive from a stratigraphic sequence similar to those observed in present day subduction initiation sequences[2]. Experiments have shown that partial melting of those rocks can reproduce the associated TTG rocks[30]. Partial melting of the Jack Hills protoliths in the presence of

residual garnet provides a viable explanation for the origin of TTGs. Finally, we find no evidence for within-plate geochemical signatures or alternation between with-plate and subduction like signatures over time. The presence of a subduction signature does not prove plate tectonics operated in the early Archaean. However, our findings are entirely permissible of an onset of modern-style plate tectonics very early in Earth history consistent with recent geodynamic modelling[31] and Si and Mo isotope studies[32,33].

## Methods

**Zircons and SHRIMP age dating.** Grains for this study were extracted from one of the vials of non-magnetic zircon prepared from the original Jack Hills conglomerate sample obtained from the W74 site[34]. Two mounts were prepared with a total of approximately 50 zircons on each, along with several pieces of the CZ3 standard with a $^{206}Pb/^{238}U$ age of 564 Ma[35]. The zircons were cast in epoxy resin, ground, polished and cleaned prior to gold coating. Images were taken in both reflected and transmitted light in order to identify any imperfections or inclusions, and the mounts were then imaged in cathodoluminescence (CL). Based on these images, not all zircons were found to be suitable for analysis due to cracks and inclusions, and a total of 37 grains were analysed on mount 1 and 24 on mount 2. Initial analyses involved single-cycle runs on a SHRIMP II at Curtin University to determine the approximate age of the selected grains and to ascertain how many were Hadean. It was established that the grains were representative of the W74 zircon population and all were then re-analysed using 6-cycle runs through the mass stations following published methods[36]. A mass resolution of ~5000 was recorded during measurement of the Pb/Pb and U/Pb isotopic ratios and the latter were normalised to those measured on the standard zircon [CZ3–($^{206}Pb/^{238}U$ = 0.0914)]. The uncertainty associated with measurement of U/Pb for the standard at 1σ was 1.52% for the mount 1 session and 1.32% for the mount 2 session. The common lead correction was modelled on the composition of Broken Hill lead and the data were processed using Squid 1[37] and Isoplot 3.75[38]. The data are contained in Supplementary Data 1. Discordant zircons (>5%) and any indicating chemical evidence of alteration[39] were removed from dataset prior to further processing and interpretation.

**Trace element determinations.** Trace-element concentrations of zircons were measured using laser ablation inductively coupled plasma mass spectrometry (LA-ICP-MS) using an Agilent 7700× quadrupole ICP-MS coupled with a Photon Machines Excite Analyte 193 nm excimer laser ablation system with HelEx sample cell at Macquarie GeoAnalytical (MQGA), Macquarie University. The data were integrated near the surface to be as close as possible to where the U-Pb ages were obtained. A fluence of 7.59 J/cm² and a repetition rate of 5 Hz were adopted for the analytical conditions for the laser. A laser beam of 50 μm was used for the zircon analysis. Each measurement consists of 60 s of background and 120 s of ablation. NIST-610 glass standards were analysed at the beginning and the end of each analysis series to correct the machine drift. Concentrations for the trace elements were obtained through calibration of relative element sensitivities using the NIST-610 standard as the external calibration standard and Zr content was used for internal calibration. Basaltic reference material, BCR-2G, and zircon reference materials, GJ and 91500, were run before the unknown samples to monitor the accuracy and reproducibility of the measurements. Data reduction was carried out online using the GLITTER software[40]. The spectra were trimmed to avoid both surface contamination and clear depth related zoning although some unavoidable variation in U and Th concentrations remains. P and REE data were then used to identify and discard any zircons containing evidence for alteration[39]. Trace element concentrations of BCR-2G and 91500 obtained in this study are in good agreement with recommended values[41,42]. The data are contained in Supplementary Data 2. Errors for calculated melt composition trace element ratios have been propagated from the external reproducibility of the laser ablation analyses and incorporate the uncertainties on the partition coefficients[22].

**Calculation of protolith $SiO_2$ contents.** In order to derive the $SiO_2$ contents of the equilibrium melts we regressed 41186 analyses between 50 and 70 wt.% $SiO_2$ from an initial data set of >100,000 whole rock obtained from the GEOROC database and unfiltered for age (see below). The data was binned into 1 wt.% $SiO_2$ intervals and a linear regression was used to determine the form $Th/Y = 0.0269 \times SiO_2–1.3169$ with an $R^2$ value of 0.9812. Since both Th and Y are compatible in zircon with $D_{zircon/melt} = 1.1$ and 2.1, respectively[22] and neither have redox sensitive partitioning their inferred melt compositions can be regarded as robust. The standard error about the mean was then subsequently calculated for each 1% bin interval. Each bin contained a minimum of $n = 527$ (66 wt.% $SiO_2$) to a maximum of 4936 (50 wt.% $SiO_2$) analyses. Further regressions were then conducted on the maximum ($Th/Y = 0.0263 \times SiO_2–1.3066$, $R^2 = 0.9812$) and minimum ($Th/Y = 0.0257 \times SiO_2–1.2963$; $R^2 = 0.9808$) values for each bin interval in order to calculate the maximum variation of $SiO_2$ within individual bin intervals. This corresponded to ±1.1 wt.% $SiO_2$ at 69 wt.% $SiO_2$ and was a minimum of 0.75 at 51 wt.% $SiO_2$

(Supplementary Fig. 1). Hence we conservatively report the purely statistical errors on the calculated $SiO_2$ to be ±1.1 wt.% $SiO_2$ (1 SE) and when this is combined with the analytical errors from the trace element analyses and errors for the partition coefficients the combined errors on calculated melt $SiO_2$ is ±3.5 wt.%. The errors on calculated $SiO_2$ are significantly less than the overall calculated variation in $SiO_2$ and thus do not impinge upon the major conclusions drawn from the calculations.

As noted above, the dataset used for the regression was unfiltered for age. If we use only Archaean magmatic rocks we obtain a regression that is $Th/Y = 0.1 \times SiO_2–6$. This leads to even higher inferred $SiO_2$ contents for the Jack Hills protoliths for 2 reasons we preferred to retain the original calculations. First, our original regression gives a "conservative" estimate of protolith $SiO_2$. Second, The Archaean-only regression is naturally dominated by TTGs yet our Fig. 2b indicates that TTGs at not an appropriate the source of the Jack Hills zircons.

## Data availability

All data used in this paper are available in Supplementary Data 1 and Supplementary Data 2.

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

## Acknowledgements
S.T. thanks the Akademie der Wissenschaften zu Göttingen for a Gauss Professorship during which the paper was written. We thank Craig O'Neill and Chris Hawkesworth for stimulating discussions about the onset of plate tectonics.

## Author contributions
S.W. selected the zircons and obtained the U-Pb age data, S.T. obtained the trace element data with Y.-J. Lai. S.T., G.W. and B.S. calculated and interpreted the equilibrium melt compositions and all authors contributed to the writing of the paper.

## Competing interests
The authors declare no competing interests.
