## [Peer Review File · Nature Communications]

Reviewers' comments:

Reviewer #1 (Remarks to the Author):

Turner and coauthors present new zircon U-Pb age and complementary trace element analyses for 61 Hadean Jack Hills zircons. These results are interpreted in the context of zircon mineral-melt partition coefficients to estimate the composition of the magma from which these zircons crystallized. They go on to propose an andesitic source magma for the Hadean zircons, suggesting a subduction-like setting throughout the Hadean. These results are certainly novel, and are likely to be of wide interest.

The discussion of partition coefficients and the caveats thereof is appropriately thorough. In their interpretations, the authors rely primarily on mineral/melt partition coefficient ratios, which, as the authors correctly note are generally more robust than absolute mineral/melt partition coefficients.

My one outstanding point of concern involves the calibration of the relationship between SiO₂ and Th/Y: by drawing on GEOROC as a whole, this calibration curve is presently dominated by relatively young Phanerozoic and Proterozoic whole-rock analyses, and would not hold exactly in the Hadean due to greater mantle potential temperature in the early Earth. This type of issue could be partially ameliorated by using only Archean analyses from GEOROC when fitting the line in Supp. Figure 1. However, it is important to note that such a correction would lead to lower Th/Y at a given SiO₂, and thus somewhat higher inferred SiO₂ for the Jack Hills zircon source magma. This, however, would likely only strengthen the authors' interpretations regarding the origins of this magma.

Any study of the Jack Hills zircons invites a wide range of entrenched opinions, and consequently such work is always likely to be controversial. From my perspective, however, this study is entirely consistent with a range of emerging evidence regarding the state of the Hadean Earth. Overall, I find that both the new dataset and the interpretation thereof represent valuable contributions to the literature and I hope to see them in print.

Other specific comments:

15: "by" is unnecessary. For that matter, I would probably avoid the claim of novelty altogether (can leave that to the cover letters) and just say "Here we use partition coefficient ratios to ... "

22-23: "In contrast to some recent" suggestions/studies would be more idiomatic. Maybe cite which ones, if this is referring to specific published papers?

54: missing comma after "heterogeneous". May want to further rephrase this sentence anyways though since "rendering ... with significant potential errors" does not flow well (c.f. "rendering ... significantly inaccurate", which would be more grammatically correct).

73-74: This is quite interesting indeed!

82: missing "or": "in or above"

252: Shouldn't that be ">100,000"?

319, 331: References to "Methods" have been lower case until now. I suspect capitalized like this is more in line with journal guidelines, but best to be consistent either way.

Fig. S1 is missing axis labels (Th/Y vs SiO₂)

Reviewer #2 (Remarks to the Author):

I may not be the best candidate to review this communication because I do not have a hands-on knowledge of the approach taken by the authors in estimating the composition of the melts from which the Jack Hills zircons crystallized. As a result, I can't evaluate if these composition estimates are robust or how dependent they are on the choice of partition coefficients. I defer to others with expertise in this field to evaluate the approach and results used in this paper.

Beyond that, there are two main aspects of the paper that I do not find terribly compelling.

First, I have little faith in using discriminant diagrams to definitively identify tectonic settings—particularly for the early Earth where the geodynamic settings are not well constrained. Can we confidently use compositions estimated from zircons in the approach employed here and compare these to modern rocks to determine their tectonic setting? I am skeptical of this comparative approach because the early Earth was fundamentally different than today from many perspectives. This paper does little to lessen this skepticism.

Second, even if there is evidence of a subduction zone signature in these estimated parental magmas, that is not the same as providing evidence for plate tectonics. It is certainly possible, if not likely, that some form of crustal "drips" into the mantle, possibly with some subduction characteristics, existed in the early Earth that was isolated and did not involve interconnected rigid plates over the surface of the Earth as exist today. In other words, a subduction zone signature is one thing; evidence of plate tectonics is another thing altogether and it is important not to equate the two

In other matters, the paper is well written and structured. Figure 2 is fine; Fig 1 could be improved. Some very minor comments (identified by line number) are given below.

Line 12-13: Zircon Hf isotope data are not really discussed in any significant way in this paper (including what these data tell us or how used in the literature) so why mention it here?

Lines 17-19: Suggest: "Using this approach, the average SiO₂ content of these melts was 59 ± 6 wt. % with Th/Nb, Dy/Yb and Sr/Y ratios of 2.7 ± 1.9 , 0.9 ± 0.2 and 1.6 ± 0.7 , respectively.

Line 48: The technique described here does not obtain these data from the "same spot". This may seem like splitting hairs but this is important in complex zircons like the JH zircons. Better to say "same location".

Line 55-57: Say with the study is first, then note that this is the "only experimental study that..."

Line 93-95: Nb doesn't have analytical difficulties—we have analytical difficulties measuring Nb

Line 118: Other refs for the 3.8 OR 4.3 Gyr age (other than O'Neil et al.)? Also it should be "or", not "to" which signifies a full range, which has not been suggested to my knowledge.

Responses to reviews (in italics and Times below)

Reviewer #1 (Remarks to the Author):

Turner and coauthors present new zircon U-Pb age and complementary trace element analyses for 61 Hadean Jack Hills zircons. These results are interpreted in the context of zircon mineral-melt partition coefficients to estimate the composition of the magma from which these zircons crystallized. They go on to propose an andesitic source magma for the Hadean zircons, suggesting a subduction-like setting throughout the Hadean. These results are certainly novel, and are likely to be of wide interest.

The discussion of partition coefficients and the caveats thereof is appropriately thorough. In their interpretations, the authors rely primarily on mineral/melt partition coefficient ratios, which, as the authors correctly note are generally more robust than absolute mineral/melt partition coefficients.

My one outstanding point of concern involves the calibration of the relationship between SiO₂ and Th/Y: by drawing on GEOROC as a whole, this calibration curve is presently dominated by relatively young Phanerozoic and Proterozoic whole-rock analyses, and would not hold exactly in the Hadean due to greater mantle potential temperature in the early Earth. This type of issue could be partially ameliorated by using only Archean analyses from GEOROC when fitting the line in Supp. Figure 1. However, it is important to note that such a correction would lead to lower Th/Y at a given SiO₂, and thus somewhat higher inferred SiO₂ for the Jack Hills zircon source magma. This, however, would likely only strengthen the authors' interpretations regarding the origins of this magma.

This is an interesting point and a regression through just Archean data yields $Th/Y = -6 + 0.1 \times SiO_2$. As the reviewer notes this leads to even higher inferred SiO₂ for the Jack Hills protolith. We have added a comment about this in the methods section but for 2 reasons prefer to retain the original calculations in the main body of the paper. First, our regression gives a "conservative" estimate of protolith SiO₂. Second, The Archean regression is dominated by TTG's yet as discussed in the text our Fig. 2B indicates that TTGs are NOT the source of the Jack Hills zircons and so using a correlation dominated by Archean TTGs seems inappropriate.

Any study of the Jack Hills zircons invites a wide range of entrenched opinions, and consequently such work is always likely to be controversial. From my perspective, however, this study is entirely consistent with a range of emerging evidence regarding the state of the Hadean Earth. Overall, I find that both the new dataset and the interpretation thereof represent valuable contributions to the literature and I hope to see them in print.

We have added two more references (31 and 32) at the end of the discussion (one based on Si isotopes and one on Mo isotopes) that support our interpretation that there was both subduction and significant amounts of silicic crust in the Hadean.

Other specific comments:

15: "by" is unnecessary. For that matter, I would probably avoid the claim of novelty altogether (can leave that to the cover letters) and just say "Here we use partition coefficient ratios to ..."

Changed accordingly.

22-23: "In contrast to some recent" suggestions/studies would be more idiomatic. Maybe cite which ones, if this is referring to specific published papers?

Changed accordingly. The appropriate references are cited later in the text.

54: missing comma after "heterogeneous". May want to further rephrase this sentence anyways though since "rendering ... with significant potential errors" does not flow well (c.f. "rendering ... significantly inaccurate", which would be more grammatically correct).

Changed accordingly.

73-74: This is quite interesting indeed!

Unfortunately, since it is only 2 of 61 zircons it will require further work to decipher the potential meaning of these higher temperatures (and we only have 2 that are 4.3 Ga).

82: missing "or": "in or above"

Changed accordingly.

252: Shouldn't that be ">100,000"?

Changed accordingly.

319, 331: References to "Methods" have been lower case until now. I suspect capitalized like this is more in line with journal guidelines, but best to be consistent either way.

Checking the journal it looks like "Methods" are in lower case so we will stick with that unless told otherwise.

Fig. S1 is missing axis labels (Th/Y vs SiO₂)

This has been fixed.

Reviewer #2 (Remarks to the Author):

I may not be the best candidate to review this communication because I do not have a hands-on knowledge of the approach taken by the authors in estimating the composition of the melts from which the Jack Hills zircons crystallized. As a result, I can't evaluate if these composition estimates are robust or how dependent they are on the choice of partition coefficients. I defer to others with expertise in this field to evaluate the approach and results used in this paper.

Beyond that, there are two main aspects of the paper that I do not find terribly compelling. First, I have little faith in using discriminant diagrams to definitively identify tectonic settings—particularly for the early Earth where the geodynamic settings are not well constrained. Can we confidently use compositions estimated from zircons in the approach employed here and compare these to modern rocks to determine their tectonic setting? I am skeptical of this comparative approach because the early Earth was fundamentally different than today from many perspectives. This paper does little to lessen this skepticism.

We acknowledge this reviewers scepticism but this paper is not the place to reappraise the use of discrimination diagrams. Julian Pearce has been very careful about how these diagrams are used and we have added an additional reference (26) which refers to a 2019 publication which has revisited this issue including the Th/Nb – Nb/Yb diagram. We have reworded lines 79-81 to acknowledge that some caution may be required in their use. It is not clear to us what else we can do here. In the absence of an alternative approach we either have to accept the Pearce approach or not. Moreover, even if this diagram is treated with caution the elevated SiO₂ and Th/Nb signals remain valid (Fig. 1).

Second, even if there is evidence of a subduction zone signature in these estimated parental magmas, that is not the same as providing evidence for plate tectonics. It is certainly possible, if not likely, that some form of crustal "drips" into the mantle, possibly with some subduction characteristics, existed in the early Earth that was isolated and did not involve interconnected rigid plates over the surface of the Earth as exist today. In other words, a subduction zone signature is one thing; evidence of plate tectonics is another thing altogether and it is important not to equate the two.

We accept this but thought that we had been appropriately prudent by using phrases like "supports onset of plate tectonics" (lines 2-3) and 'are entirely permissible of an early onset of modern style plate tectonics" (lines 25-26). We have modified the final sentence (lines 125-126) to begin with the caveat "Whether this provides evidence for

plate tectonics may require further work but our findings...”. Moreover, at least there is no disagreement as to what a subduction signature is – whereas the signature of crustal drips (if they existed) is unconstrained. We re-emphasise this at the end of the paper (lines 128-129).

In other matters, the paper is well written and structured. Figure 2 is fine; Fig 1 could be improved. Some very minor comments (identified by line number) are given below.

Thank you! There is no indication as to what needs to be changed in Fig. 1? It looks OK to us?

Line 12-13: Zircon Hf isotope data are not really discussed in any significant way in this paper (including what these data tell us or how used in the literature) so why mention it here?

Simply to give some context to the non-specialists. Since there are so many papers on Hf isotopes in the Jack Hills zircons it seemed appropriate to point out that we are taking a different approach to a well-debated problem. Therefore, we prefer to keep this sentence.

Lines 17-19: Suggest: “Using this approach, the average SiO₂ content of these melts was 59 ± 6 wt. % with Th/Nb, Dy/Yb and Sr/Y ratios of 2.7 ± 1.9, 0.9 ± 0.2 and 1.6 ± 0.7, respectively.

Changed as suggested.

Line 48: The technique described here does not obtain these data from the “same spot”. This may seem like splitting hairs but this is important in complex zircons like the JH zircons. Better to say “same location”.

Changed as suggested.

Line 55-57: Say with the study is first, then note that this is the “only experimental study that...”

We couldn't quite understand what change was being suggested here.

Line 93-95: Nb doesn't have analytical difficulties—we have analytical difficulties measuring Nb

Indeed! We have reworded accordingly.

Line 118: Other refs for the 3.8 OR 4.3 Gyr age (other than O'Neil et al.)? Also it should be “or”, not “to” which signifies a full range, which has not been suggested to my knowledge.

We have changed “to” to “or” which is correct. However, as far as we are aware there have not been any other dating studies of the basalts from the Nuvvuagittuq Greenstone Belt. Only the tonalities have been dated by other groups.

REVIEWERS' COMMENTS:

Reviewer #1 (Remarks to the Author):

I am satisfied with the authors response to my comments, and look forward to seeing the work in print.

Reviewer #2 (Remarks to the Author):

Excerpts from the ms and the authors' responses are given below in italics (Times 10 point). My old comments are in Helvetica (9 point); my new comments are in Helvetica 11 point and indented

First, I have little faith in using discriminant diagrams to definitively identify tectonic settings—particularly for the early Earth where the geodynamic settings are not well constrained. Can we confidently use compositions estimated from zircons in the approach employed here and compare these to modern rocks to determine their tectonic setting? I am skeptical of this comparative approach because the early Earth was fundamentally different than today from many perspectives. This paper does little to lessen this skepticism.

We acknowledge this reviewers scepticism but this paper is not the place to reappraise the use of discrimination diagrams. Julian Pearce has been very careful about how these diagrams are used and we have added an additional reference (26) which refers to a 2019 publication which has revisited this issue including the Th/Nb – Nb/Yb diagram. We have reworded lines 79-81 to acknowledge that some caution may be required in their use. It is not clear to us what else we can do here. In the absence of an alternative approach we either have to accept the Pearce approach or not. Moreover, even if this diagram is treated with caution the elevated SiO₂ and Th/Nb signals remain valid (Fig. 1).

Using these discrimination diagrams is not only an exercise in stamp collecting, but attempts to match stamps from one country/era to another. Not compelling science, imo.

Second, even if there is evidence of a subduction zone signature in these estimated parental magmas, that is not the same as providing evidence for plate tectonics. It is certainly possible, if not likely, that some form of crustal "drips" into the mantle, possibly with some subduction characteristics, existed in the early Earth that was isolated and did not involve interconnected rigid plates over the surface of the Earth as exist today. In other words, a subduction zone signature is one thing; evidence of plate tectonics is another thing altogether and it is important not to equate the two.

We accept this but thought that we had been appropriately prudent by using phrases like "supports onset of plate tectonics" (lines 2-3) and 'are entirely permissible of an early onset of modern style plate tectonics" (lines 25-26). We have modified the final sentence (lines 125-126) to begin with the caveat "Whether this provides evidence for plate tectonics may require further work but our findings...". Moreover, at least there is no disagreement as to what a subduction signature is – whereas the signature of crustal drips (if they existed) is unconstrained. We re-emphasise this at the end of the paper (lines 128-129).

I. 25-26. These results are entirely permissible of an early onset of modern-style plate tectonics on Earth.

It is essential to separate "subduction zone signature" from early onset of plate tectonics. Full stop. You could certainly have the former without the latter. There are many papers that propose modern style plate-tectonics based on equivocal trace-element data. There is no reason to add another. Conjecture like this takes away from the credibility of the paper, in my opinion, because it appears that the authors have an agenda that they are trying to justify. If the results indicate a subduction zone signature then make that statement and leave it at that. The information presented here certainly doesn't provide evidence for the existence of interconnected plates interacting across the entire surface of the planet such as what we have with modern plate

tectonics.

l. 11-13. The composition and origin of Earth's early crust remains hotly debated with many arguments based on the Hf isotope composition of Archaean detrital zircons from Jack Hills in Western Australia

l. 8-9. Many of these conclusions are based on the Hf isotope signature of the zircons

This is not true. In my opinion, Hf isotopes can't be used to make most of the statements that precede this statement. Other information from zircon (e.g., O-isotopes, trace-elements) has been used more for this purpose. If the authors want to make this point (composition and origin of Earth's early crust...based on the Hf isotope composition of detrital zircons) they need to state what zircon Hf isotope evidence has used to support these claims. I have no clue what Hf isotope data the authors are talking about.

l. 75. However, before drawing conclusions on this finding, more measurements would be needed. This is an empty statement. Delete.

l. 81-2. their use, plots like the Th/Yb versus Nb/Yb diagram shown in Figure 2A have been repeatedly appraised to help distinguish the tectonic setting in which ancient rocks formed remove "repeatedly"

l. 119-121. Interestingly the average composition of the inferred Jack Hills protolith is very similar to the average low-Ti enriched basalts from the Nuvvuagittuq Greenstone Belt ($\text{SiO}_2 \sim 55 \text{ wt. } \%$; $\text{Th/Nb} \sim 0.8$) that may be 3.8 or 4.3 Gyr in age.

I am confused. I thought the authors state that the inferred equilibrium melts are much more similar to andesites formed in modern subduction settings. The low-Ti enriched basalt from Nuvvuagittuq is a different beast entirely, isn't it?

Line 55-57: Say with the study is first, then note that this is the "only experimental study that..." We couldn't quite understand what change was being suggested here.

l. 55-57. Accordingly, we have used LA-ICP-MS partition coefficients from the only experimental study that analysed both a wide range of trace elements and that also showed these to be consistent with the lattice strain model.

My comment meant to say "state what that experimental study is first, then note that this is the "only experimental study that...". In other words, provide some context to this statement.

Responses to additional comments from reviewer #2

Using these discrimination diagrams is not only an exercise in stamp collecting, but attempts to match stamps from one country/era to another. Not compelling science, imo.

We do not know how to respond to this more than we have already. Clearly this reviewer and we have different opinions. The vast number of citations of the papers by Julian Pearce suggests to us that they have been generally accepted by the community. Both the elevated SiO₂ and Th/Nb are arc-like signals even if Julians diagram is completely ignored.

It is essential to separate “subduction zone signature” from early onset of plate tectonics. Full stop. You could certainly have the former without the latter. There are many papers that propose modern style plate-tectonics based on equivocal trace-element data. There is no reason to add another. Conjecture like this takes away from the credibility of the paper, in my opinion, because it appears that the authors have an agenda that they are trying to justify. If the results indicate a subduction zone signature then make that statement and leave it at that. The information presented here certainly doesn't provide evidence for the existence of interconnected plates interacting across the entire surface of the planet such as what we have with modern plate tectonics.

Surely we have done this! Lines 130-133 state: “The presence of a subduction signature does not prove plate tectonics operated in the early Archaean. However, our findings are entirely permissible of an onset of modern-style plate tectonics very early in Earth history consistent with recent geodynamic modelling³¹ and Si and Mo isotope studies^{32,33}.” We are clearly not going to change this reviewers view point (who clearly has an agenda too) but being a little provocative will at least generate discussion.

This is not true. In my opinion, Hf isotopes can't be used to make most of the statements that precede this statement. Other information from zircon (e.g., O-isotopes, trace-elements) has been used more for this purpose. If the authors want to make this point (composition and origin of Earth's early crust...based on the Hf isotope composition of detrital zircons) they need to state what zircon Hf isotope evidence has used to support these claims. I have no clue what Hf isotope data the authors are talking about.

This has been reworded as such “Some inferences about the bulk composition of their protoliths have come from the Hf isotope composition of the zircons^{5,8,9}.” because Hf isotopes have been a major source of information for inferring a mafic rather than intermediate (as we infer) bulk composition of the protoliths.

I am confused. I thought the authors state that the inferred equilibrium melts are much more similar to andesites formed in modern subduction settings. The low-Ti enriched basalt from Nuvvuagittuq is a different beast entirely, isn't it?

This is incorrect. The average SiO₂ content of the low-Ti enriched basalts is 53.5% which is a basaltic andesite. We made an error in calling them basalts in the previous version and line 124 now clarifies that they are basaltic andesites.

My comment meant to say “state what that experimental study is first, then note that this is the “only experimental study that...”. In other words, provide some context to this statement.

This has been changed accordingly.